# PREDICTIVE, SCALABLE AND INTERPRETABLE KNOWLEDGE TRACING ON STRUCTURED DOMAINS

**Hanqi Zhou**[1,2,4]**, Robert Bamler**[1,3]**, Charley M. Wu**[1,2,3]*,**& Álvaro Tejero-Cantero**[1,2]*

[1]University of Tübingen, [2]Cluster of Excellence Machine Learning, [3]Tübingen AI Center, [4]IMPRS-IS
{hanqi.zhou,robert.bamler,charley.wu,alvaro.tejero}@uni-tuebingen.de

## ABSTRACT

Intelligent tutoring systems optimize the selection and timing of learning materials to enhance understanding and long-term retention. This requires estimates of both the learner's progress ("knowledge tracing"; KT), and the prerequisite structure of the learning domain ("knowledge mapping"). While recent deep learning models achieve high KT accuracy, they do so at the expense of the interpretability of psychologically-inspired models. In this work, we present a solution to this trade-off. PSI-KT is a hierarchical generative approach that explicitly models how both individual cognitive traits and the prerequisite structure of knowledge influence learning dynamics, thus achieving interpretability by design. Moreover, by using scalable Bayesian inference, PSI-KT targets the real-world need for efficient personalization even with a growing body of learners and learning histories. Evaluated on three datasets from online learning platforms, PSI-KT achieves superior multi-step **p**redictive accuracy and **s**calable inference in continual-learning settings, all while providing **i**nterpretable representations of learner-specific traits and the prerequisite structure of knowledge that causally supports learning. In sum, predictive, scalable and interpretable knowledge tracing with solid knowledge mapping lays a key foundation for effective personalized learning to make education accessible to a broad, global audience.

## 1 INTRODUCTION

The rise of online education platforms has created new opportunities for personalization in learning, motivating a renewed interest in how humans learn structured knowledge domains. Foundational theories in psychology (Ebbinghaus, 1885) have informed *spaced repetition* schedules (Settles & Meeder, 2016), which exploit the finding that an optimal spacing of learning sessions enhances memory retention. Yet beyond the timing of rehearsals, the sequential order of learning materials is also crucial, as evidenced by curriculum effects in learning (Dewey, 1910; Dekker et al., 2022), where exposure to simpler, prerequisite concepts can facilitate the apprehension of higher-level ideas. Cognitive science and pedagogical theories have long emphasized the relational structure of knowledge in human learning (Rumelhart, 2017; Piaget, 1970), with recent research showing that mastering prerequisites enhances concept learning (Lynn & Bassett, 2020; Karuza et al., 2016; Brändle et al., 2022). Yet, we still lack a predictive, scalable, and interpretable model of the structural-temporal dynamics of learning that could be used to develop future intelligent tutoring systems.

Here, we present PSI-KT, a novel approach for inferring interpretable learner-specific cognitive traits and a shared knowledge graph of prerequisite concepts. We demonstrate our approach on three real-world educational datasets covering structured domains, where our model outperforms existing baselines in terms of *predictive* accuracy (both within- and between-learner generalization), *scalability* in a continual learning setting, and *interpretability* of learner traits and prerequisite graphs. The paper is organized as follows: We first introduce the knowledge tracing problem and summarize related work (Sec. 2). We then provide a formal description of PSI-KT and describe the inference method (Sec. 3). Experimental evaluations are organized into demonstrations of prediction performance, scalability, and interpretability (Sec. 4). Altogether, PSI-KT bridges machine learning and cognitive science, leveraging our understanding of human learning to build the foundations for automated tutoring systems with broad educational applications.

---

*Equal contribution. Code at github.com/mlcolab/psi-kt

## 2 BACKGROUND

In this section, we begin by defining the knowledge tracing problem and then review related work.

### 2.1 KNOWLEDGE TRACING FOR INTELLIGENT TUTORING SYSTEMS

For almost 100 years (Pressey, 1926), researchers have developed intelligent tutoring systems (ITS) to support human learning through adaptive teaching materials and feedback. More recently, *knowledge tracing* (KT; Corbett & Anderson, 1994) emerged as a method for tracking learning progress by predicting a learner's performance on different *knowledge components* (KCs), e.g., the 'Pythagorean theorem', based on past learning interactions. Here, we focus on the KT problem, with the goal of supporting the selection of teaching materials in future ITS applications.

In this setting, a learner $\ell$ receives exercises or flashcards for KCs $x_n^\ell \in \{0, 1, \ldots, K\}$ at irregularly spaced times $t_n^\ell$, whereupon the performance is recorded, often as correct/incorrect, $y_n^\ell \in \{0, 1\}$. We can formalize KT as a supervised learning problem on time-series data, where the goal of the KT model is to predict future performance (e.g., $\hat{y}_{N+1}$) given all or part of the interaction history $\mathcal{H}_{1:N}^\ell := \{x_n^\ell, t_n^\ell, y_n^\ell\}_{n=1}^N$ available up to time $t_N^\ell$. As part of the process, a KT model may infer specific representations of learners or of the learning domain to help prediction. If these representations are interpretable, they can be valuable for downstream learning personalization.

### 2.2 RELATED WORK

We broadly categorize related KT approaches into psychological and deep learning methods.

**Psychological methods.** Focusing on interpretability, psychological methods use domain knowledge to describe the temporal decay of memory (e.g., forgetting curves; Ebbinghaus, 1885), sometimes also modeling learner-specific characteristics. *Factor-based regression* models use hand-crafted features based on learner interactions and KC properties (e.g., repetition counts and KC easiness; Pavlik Jr et al., 2009). While they model KC-dependent memory dynamics (Pavlik et al., 2021; Gervet et al., 2020; Lindsey et al., 2014; Lord, 2012; Ackerman, 2014), they ignore the relational structure between KCs. Half-life Regression (HLR; Settles & Meeder, 2016) from Duolingo uses both correct and incorrect counts, while the Predictive Performance Equation (PPE; Walsh et al., 2018) models the elapsed time of every past interaction with a power function to account for spacing effects. By using shallow regression models with predefined features, these models achieve interpretability, but sacrifice prediction accuracy. *Latent variable models* use a probabilistic two-state Hidden Markov Model (Käser et al., 2017; Sao Pedro et al., 2013; Baker et al., 2008; Yudelson et al., 2013), representing either mastery or non-mastery of a given KC. These models are limited to binary states by design, do not account for learner dynamics, and for some, their numerous parameters hinder scalability. Another probabilistic model, HKT (Wang et al., 2021) accounts for structure and dynamics by modeling knowledge evolution as a multivariate Hawkes process. Close in spirit to our PSI-KT, this approach tracks KC structure but lacks any learner-specific representations.

**Deep learning methods.** Deep learning methods use flexible models with many parameters to achieve high prediction accuracy. However, this flexibility also makes it difficult to interpret their learned internal representations. The first deep learning methods explicitly modeled sequential interactions with *recurrent neural networks* to overcome the dependence on fixed summary statistics in simpler regression models, with Deep Knowledge Tracing (DKT; Piech et al., 2015) pioneering the use of Long Short-Term Memory (LSTM) networks (Hochreiter & Schmidhuber, 1997). A similar architecture, DKTF (Nagatani et al., 2019) incorporated additional input features, whereas Shen et al. (2021) proposed an intricate modular architecture aimed at recovering interpretable learner representations, but neglecting KC relations. *Structure-aware models* leverage KC dependencies, accounting for the fact that human knowledge acquisition is structured by dependency relationships (i.e., concept maps; Hill, 2005; Koponen & Nousiainen, 2018; Lynn & Bassett, 2020). Tong et al. (2020) empirically estimate KC dependencies from the frequencies of successful transitions. AKT (Ghosh et al., 2020) relies on the attention mechanism (Vaswani et al., 2017) to implicitly capture structure (Pandey & Karypis, 2019; Choi et al., 2020; Shin et al., 2021; Liu et al., 2023), whereas GKT (Nakagawa et al., 2019) models it explicitly based on graph neural networks (Kipf & Welling, 2016). Recent work towards interpretable deep learning KT uses engineered features such as learner mastery and exercise difficulty (Minn et al., 2022), or infers them with neural networks (QIKT; Chen et al., 2023, IEKT; Long et al., 2021). While diverse approaches to interpretability exist (see Chen et al., 2023, for review), a comprehensive evaluation framework is still lacking.

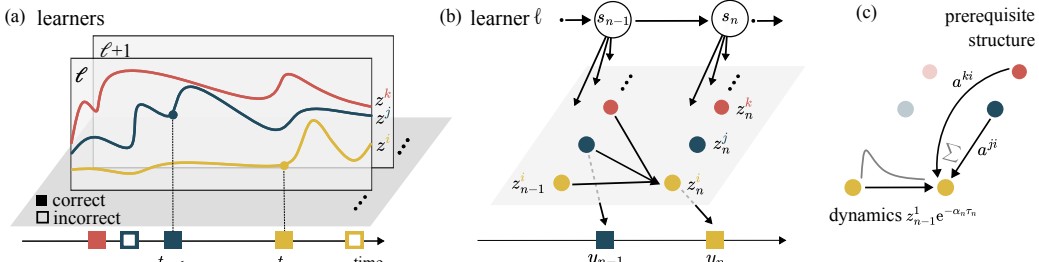

Figure 1: PSI-KT is a hierarchical probabilistic state-space model of learning. *(a)* Latent knowledge states for different KCs (colored curves) are inferred from observations. *(b)* Full hierarchical model for a single learner: cognitive traits $s_n$ control the coupled dynamics of states $z_n^k$, which give rise to observations $y_n$. *(c)* The dynamics combine memory decay (Eq. 6) and structural influences (Eq. 5).

Here, we present our predictive, scalable and interpretable KT model (PSI-KT) as a psychologically-informed probabilistic deep learning approach, together with a comprehensive evaluation framework for interpretability.

## 3 JOINT DYNAMICAL AND STRUCTURAL MODEL OF LEARNING

In this section, we describe PSI-KT, our probabilistic hierarchical state-space model of human learning (Fig. 1). Briefly, observations of learner performance $y$ (Fig. 1a, filled/unfilled boxes) provide indirect and noisy evidence about latent knowledge states $z$ (colored curves, with matching dots in Fig. 1b). These latent states evolve stochastically, in line with the psychophysics of memory (temporal decay in Fig. 1c), while simultaneously being subject to structural influences from performance on prerequisite KCs (structure in Fig. 1c). We introduce a second latent level of learner-specific traits $s$ (Fig. 1b, top), which govern the knowledge dynamics in an interpretable way.

Below, we describe the method in more detail. We start with the generative model (Sec. 3.1). Next, we discuss the joint approximate Bayesian inference of latent variables and estimation of generative parameters (Sec. 3.2). Finally, we show how to derive multi-step performance predictions (see Sec. 3.3 and Fig. 7 in Appendix A.4 for a graphical overview of inference and prediction).

### 3.1 PROBABILISTIC STATE-SPACE GENERATIVE MODEL

We conceptualize observations of learner performance as noisy measurements of an underlying time-dependent knowledge state, specific to each learner and KC. The evolution of knowledge states reflects the process of learning and forgetting, governed by learner-specific traits. Additionally, knowledge of different KCs informs one another according to learned prerequisite relationships. We translate these modeling assumptions into a generative model consisting of three main components:(i) the learner knowledge state across KCs, $\boldsymbol{z}_n^\ell = [z_n^{\ell,1} \dots z_n^{\ell,K}]^\mathsf{T} \in \mathbb{R}^K$ (colored curves in Fig. 1a), (ii) learner-specific cognitive traits $s_n^\ell \in \mathbb{R}^4$ (top row in Fig. 1b), and (iii) a shared static graph $\mathcal{A}$ of KCs whose edges $a^{ik}$ quantify the probability for a KC $i$ to be a prerequisite for KC $k$ (Fig. 1c).

**State-space model.** State-space models (SSMs) are a framework for partially observable dynamical processes. They represent the inherent noise of measurements $y$ by an emission distribution $p(y_n \,|\, z_n)$, separate from the stochasticity of state dynamics, modeled as a first-order Markov process with transition probabilities $p(z_n \,|\, z_{n-1})$. The state dynamics are initiated by sampling from an initial prior $p(z_1)$ to iteratively feed the transition kernel, and predictions can be drawn at any time from the emission distribution. To represent the influence of individual cognitive traits over the knowledge dynamics, we additionally condition the $z$-transitions on the traits $s$ (which also can be observed only indirectly). The three-level SSM hierarchy of PSI-KT consists of:

Level 2 (latent cognitive traits): $\qquad s_n^\ell \sim p_\theta(s_n^\ell \,|\, s_{n-1}^\ell) := \mathcal{N}(s_n^\ell \,|\, H s_{n-1}^\ell, R)$ (1)

Level 1 (latent knowledge states): $\qquad \boldsymbol{z}_n^\ell \sim p_\theta(\boldsymbol{z}_n^\ell \,|\, \boldsymbol{z}_{n-1}^\ell, s_n^\ell) := \prod_k \mathcal{N}(z_n^{\ell,k} | m_n^{\ell,k}, w_n^\ell)$ (2)

Level 0 (observed learner performance): $\quad \hat{y}_n^\ell \sim p(y_n^\ell \,|\, z_n^{\ell,k}) := \mathrm{Bern}(\mathrm{sigmoid}(z_n^{\ell,k}))$. (3)

The choice of Gaussian initial priors (discussed below) and Gaussian transitions ensures tractability, while the Bernoulli emissions model the observed binary outcomes. We now unpack this model and all its parameters in detail, starting with the knowledge dynamics.

**Knowledge states $z$.**   Recent KT methods (e.g., Nagatani et al., 2019) use an exponential forgetting function based on psychological theories (Ebbinghaus, 1885). Here, we augment this approach by adding stable long-term memory (Averell & Heathcote, 2011), and model the knowledge dynamics $z^{\ell,k}$ of an isolated KC $k$ as a mean-reverting stochastic (Ornstein-Uhlenbeck; OU) process:

$$\mathrm{d}z^{\ell,k}/\mathrm{d}t = \alpha^\ell(\mu^\ell - z^{\ell,k}) + \sigma^\ell \eta(t). \tag{4}$$

Accordingly, the state of knowledge $z^\ell$ gradually reverts to a long-term mean $\mu^\ell$ with rate $\alpha^\ell$, subject to white noise fluctuations $\eta(t)$ scaled by volatility $\sigma^\ell$. To account for the influence of other KCs, we adjust the mean $\mu_n^\ell$ using prerequisite weights $a^{ik}$ (defined in Eq. 7 below), modulated by the learner's transfer ability $\gamma_n^\ell$:

$$\tilde{\mu}_n^{\ell,k} := \mu_n^\ell + (\gamma_n^\ell/K) \sum_{i \neq k} a^{ik} z_n^{\ell,i}. \tag{5}$$

We obtain the mean $m_n^{\ell,k}$ and variance $w_n^\ell$ of the transition kernel in Eq. 2 by marginalizing the OU process over one time step $\tau_n^\ell := t_n^\ell - t_{n-1}^\ell$, which can be done analytically[1] ,

$$m_n^{\ell,k} = r_n^\ell z_{n-1}^{\ell,k} + (1 - r_n^\ell)\tilde{\mu}_n^{\ell,k}, \quad \text{with retention ratio } r_n^\ell := \mathrm{e}^{-\alpha_n^\ell \tau_n^\ell} \in (0,1). \tag{6}$$

As the time since the last interaction $\tau_n^\ell$ grows, the retention ratio $r_n^\ell$ decreases exponentially with rate $\alpha_n^\ell$, and the knowledge state reverts to the long-term mean $\tilde{\mu}_n^{\ell,k}$, which partly depends on the learner's mastery of prerequisite KCs (Eq. 5). This balances short-term and long-term learning, reflecting empirical findings from memory research (Averell & Heathcote, 2011). The structural influences are accounted for in the dynamics of $z_n^{\ell,k}$, thus justifying the conditional independence assumed in Eq. 2. A Gaussian initial prior $p_\theta(z_1^{\ell,k}) = \mathcal{N}(z_1^{\ell,k}|\bar{z}, w_1)$, where $\bar{z}, w_1 \in \mathbb{R}$ are part of the generative parameters $\theta$, completes our dynamical model of knowledge states.

**Learner-specific cognitive traits $s$.**   The dynamics of knowledge states (Eqs. 4- 6) are parameterized by learner-specific cognitive traits $(\alpha_n^\ell, \mu_n^\ell, \sigma_n^\ell, \gamma_n^\ell)$, which we collectively denote $s_n^\ell$. Specifically, $\alpha^\ell$ represents the forgetting rate (Ebbinghaus, 1885; Averell & Heathcote, 2011), $\mu^\ell$ (via $\tilde{\mu}_k^{\ell,n}$) captures long-term memory consolidation (Meeter & Murre, 2004) for practiced KCs and expected performance for novel KCs, $\sigma^\ell$ quantifies knowledge volatility, and $\gamma^\ell$ measures transfer ability (Bassett & Mattar, 2017) from knowledge of prerequisite KCs. These traits can develop during learning according to Eq. 1, starting from a Gaussian prior $p_\theta(s_1^\ell) = \mathcal{N}(s_1^\ell|\bar{s}, R_1)$ where $\bar{s} \in \mathbb{R}^4$ and the diagonal matrices $H, R_1, R \in \mathbb{R}^{4 \times 4}$ are also part of the global parameters $\theta$.

**Shared prerequisite graph $\mathcal{A}$.**   In our model, prerequisite relations influence knowledge dynamics via the coupling introduced in Eq. 5. We now discuss an appropriate parameterization for the weight matrix of the prerequisite graph, $\mathcal{A} := \{a^{ik}\}_{i,k \in 1:K}$. We assume that prerequisites are time- and learner-independent so that, in the spirit of collaborative filtering (Breese et al., 2013), we can pool evidence from all learners to estimate them. To prevent a quadratic scaling in the number of KCs, we do not directly model edge weights but derive them from KC embedding vectors $u^k$ in lower dimension $u^k \in \mathbb{R}^D$ with $D \ll K$, collected in embedding matrix $U_{K \times D}$. A basic integrity constraint for a connected pair is that dependence of KC $i$ on KC $k$ should trade off against that of $k$ on $i$, i.e., no mutual prerequisites: $a^{ik} + a^{ki} = 1$. With this in mind, we exploit the factorization of $a^{ik}$ introduced by Lippe et al. (2021) in terms of a separate probability of edge existence $p(i \multimap k)$ and definite directionality $p(i \rightarrow k \,|\, i \multimap k)$:

$$a^{ik} := p(i \rightarrow k \,|\, i \multimap k)\, p(i \multimap k)$$
$$= \mathrm{sigmoid}((u^i)^\intercal u^k)\, \mathrm{sigmoid}((u^i)^\intercal (M - M^\intercal) u^k), \tag{7}$$

where the skew-symmetric combination $M - M^\intercal$ of a learnable matrix $M$ prevents mutual prerequisites. Having presented the generative model, we now turn to inference and prediction.

## 3.2   APPROXIMATE BAYESIAN INFERENCE AND AMORTIZATION WITH A NEURAL NETWORK

We now describe how we learn the generative model parameters $\theta$ and how we infer the latent states $s, z$ introduced in Section 3.1 using a neural network ("inference network"). Since learner-specific latent states $s$ and $z$ are deducible solely from limited individual data, we expect non-negligible uncertainty. This motivates our probabilistic treatment of these states using approximate Bayesian inference. By contrast, the model parameters $\theta$ (KC parameters $U, M$ in Eq. 7, transition parameters $\bar{s}, H, R_1, R$ in Eq. 1, and $\bar{z}, w_1$ in Eq. 2) can be estimated from all learners, and we thus

---

[1] Särkkä & Solin (2019) — the variance is $w_n^\ell = (\sigma_n^\ell)^2(1 - \mathrm{e}^{-2\alpha_n^\ell \tau_n^\ell})/(2\alpha_n^\ell)$.

treat them as point-estimated parameters as described below (detailed derivation in Appendix A.1.) Here, without loss of generality, we show the inference for a single learner.

### 3.2.1 INFERENCE ON A FIXED LEARNING HISTORY

Here, we assume the full interaction history $\mathcal{H}_{1:N}^\ell$ is available for inferring the posterior over latents $p_\theta(z_{1:N}^\ell, s_{1:N}^\ell \mid y_{1:N}^\ell)$. We approach the problem using variational inference (VI). In VI, we select a distribution family $q_\phi$ with free parameters $\phi$ to approximate the posterior $p_\theta$ by minimizing their Kullback-Leibler divergence. This can only be done indirectly, by maximizing a lower bound to the marginal probability of the data, the *evidence lower bound* (ELBO). Here, we adopt the mean-field approximation $q_\phi(z_{1:N}^\ell, s_{1:N}^\ell \mid y_{1:N}^\ell) = q_\phi(z_{1:N}^\ell)\, q_\phi(s_{1:N}^\ell)$ and jointly optimize the generative $\theta$ and variational $\phi$ parameters using variational *expectation maximization* (EM; Dempster et al., 1977; Beal & Ghahramani, 2003; Attias, 1999). Motivated by real-world scalability, we introduce an inference network (see Appendix A.3 for the architecture) to amortize the learning of variational parameters $\phi$ across learners, and we employ the *reparametrization trick* (Kingma & Welling, 2014) to optimize the single-learner ELBO:

$$\text{ELBO}^\ell(\theta, \phi) = \mathbb{E}_{q_\phi(s_{1:N}^\ell)}\big[ -\log q_\phi(s_{1:N}^\ell) + \log p_\theta(s_1^\ell) + \sum_{n=2}^N \log p_\theta(s_n^\ell \mid s_{n-1}^\ell) \big]$$
$$+ \mathbb{E}_{q_\phi(z_{1:N}^\ell)}\big[ -\log q_\phi(z_{1:N}^\ell) + \log p_\theta(z_1^\ell) + \sum_{n=1}^N \log p_\theta(y_n^\ell \mid z_n^{\ell, x_n}) \big]$$
$$+ \mathbb{E}_{q_\phi(z_{1:N}^\ell)\, q_\phi(s_{1:N}^\ell)}\big[ \sum_{n=2}^N \log p_\theta(z_n^\ell \mid z_{n-1}^\ell, s_n^\ell) \big]. \tag{8}$$

The SSM emissions and transitions were introduced in Eqs. 1-3, along with the respective initial priors. To allow for a diversity of combinations of learner traits to account for the data, we model the variational posterior across learners, $q_\phi(s_{1:N})$, as a mixture of Gaussians (see Appendix A.4).

### 3.2.2 INFERENCE IN CONTINUAL LEARNING

In real-world educational settings, a KT model must flexibly adapt its current variational parameters $\phi_n$ with newly available interactions $(x_{n+1}^\ell, t_{n+1}^\ell, y_{n+1}^\ell)$. Retraining on a fixed, augmented history $\mathcal{H}_{n+1}^\ell$ to obtain an updated $\phi_{n+1}$ is possible (Eq. 8), but expensive. Instead, in PSI-KT, we use the parameters $\phi_n$ of the current posterior $q_{\phi_n}(z_n^\ell, s_n^\ell)$ to form a next-time prior,

$$\tilde{p}(z_{n+1}^\ell, s_{n+1}^\ell) := \mathbb{E}_{q_{\phi_n}(z_n^\ell, s_n^\ell \mid y_{1:n}^\ell)}\big[ p_\theta(s_{n+1}^\ell \mid s_n^\ell)\, p_\theta(z_{n+1}^\ell \mid s_{n+1}^\ell, z_n^\ell) \big]. \tag{9}$$

Due to the Bayesian nature of our model, we can now update this prior with the new evidence $y_{n+1}^\ell$ at time $t_{n+1}^\ell$ using *variational continual learning* (VCL; Nguyen et al., 2017; Loo et al., 2020), i.e., by maximizing the ELBO:

$$\text{ELBO}_{\text{VCL}}^\ell(\theta, \phi_{n+1}) = \mathbb{E}_{q_{\phi_{n+1}}(s_{n+1}^\ell)}\big[ -\log q_{\phi_{n+1}}(s_{n+1}^\ell) \big]$$
$$+ \mathbb{E}_{q_{\phi_{n+1}}(z_{n+1}^\ell)}\big[ -\log q_{\phi_{n+1}}(z_{n+1}^\ell) + \log p_\theta(y_{n+1}^\ell \mid z_{n+1}^{\ell, x_{n+1}}) \big]$$
$$+ \mathbb{E}_{q_{\phi_{n+1}}(z_{n+1}^\ell, s_{n+1}^\ell)}\big[ \log \tilde{p}(z_{n+1}^\ell, s_{n+1}^\ell) \big]. \tag{10}$$

Maximizing this $\text{ELBO}_{\text{VCL}}^\ell$ allows us to update the parameters $\phi_{n+1}$ based on a new interaction $(x_{n+1}^\ell, t_{n+1}^\ell, y_{n+1}^\ell)$ directly from the previous parameters $\phi_n$, i.e., without retraining.

### 3.3 PREDICTIONS

To predict a learner's performance on KC $x_{n+1}^\ell$ at $t_{n+1}^\ell$, we take the current variational distributions over $s_n^\ell$ and $z_n^\ell$ and transport them forward by analytically convolving them with the respective transition kernels (Eqs. 1 and 2). We then draw $z_{n+1}^{\ell, x_{n+1}}$ from the resulting distribution, and predict the outcome $\hat{y}_{n+1}^\ell$ by Eq. 3. When predicting multiple steps ahead, we repeat this procedure without conditioning on any of the previously predicted $\hat{y}_{n+m}^\ell$.

## 4 EVALUATIONS

We argue above that KT for personalized education must predict accurately, scale well with new data, and provide interpretable representations. We now empirically assess these desiderata, comparing PSI-KT with up to 8 baseline models across three datasets from online education platforms. Concretely, we evaluate (i) prediction accuracy,

Table 1: Dataset characteristics

| Dataset → | Assist12 | Assist17 | Junyi15 |
|---|---|---|---|
| # Learners $L$ | 46,674 | 1,709 | 247,606 |
| # KCs $K$ | 263 | 102 | 722 |
| # Int's / $10^6$ | 3.5 | 0.9 | 26 |

Figure 2: Within-learner prediction performance (mean $\pm$SEM) as a function of cohort sizes from 100 to the maximum available in each dataset (we omit HLR for legibility; see Table 2.)

quantifying both within-learner prediction and between-learner generalization (Sec. 4.1), (ii) scalability in a continual learning setting (Sec. 4.2), and (iii) interpretability of learner representations and prerequisite relations (Sec. 4.3).

**Datasets.** Assistments and Junyi Academy are non-profit online learning platforms for pre-college mathematics. We use Assistments' 2012 and 2017 datasets[2] (Assist12 and Assist17) and Junyi's 2015 dataset[3] (Junyi15; Chang et al., 2015), which in addition to interaction data, provides human-annotated KC relations (see Table 1 and Appendix A.3.2 for details).

We select HLR from Duolingo and PPE as two influential psychologically-informed regression models. From the models that use learnable representations, we include two established deep learning benchmarks, DKT and DKTF, which capture complex dynamics via LSTM networks, as well as the interpretability-oriented QIKT.

## 4.1 PREDICTION AND GENERALIZATION PERFORMANCE

In our evaluations, we mainly focus on prediction and generalization when training on 10 interactions from up to 1000 learners. Good KT performance with little data is key in practical ITS to minimize the number of learners on an experimental treatment (principle of equipoise, similar to medical research; Burkholder, 2021), to mitigate the cold-start problem, and to extend the usefulness of the model to classroom-size groups. To provide ITS with a basis for adaptive guidance and long-term learner assessment, we always predict the 10 next interactions. Figure 2 shows that PSI-KT's *within-learner prediction performance* is robustly above baselines for all but the largest cohorts (>60k learners, Junyi15), where all deep learning models perform similarly. The advantage of PSI-KT comes from its combined modeling of KC prerequisite relations and individual learner traits that evolve in time (see Appendix Fig. 13 for ablations). The *between-learner generalization* accuracy of the models above, when tested on 100 out-of-sample learners, is shown in Table 2, where fine-tuning indicates that parameters were updated using (10-point) learning histories from the unseen learners. PSI-KT shows overall superior generalization except on Junyi15 (when fine-tuning).

## 4.2 SCALABILITY IN CONTINUAL LEARNING

In addition to training on fixed historical data, we also conduct experiments to demonstrate PSI-KT's scalability when iteratively retraining on additional interaction data from each learner. This parallels real-world educational scenarios, where learners are continuously learning (Sec. 3.2.2). Each model is initially trained on 10 interactions from 100 learners. We then incrementally provide one data point from each learner, and evaluate the training costs and prediction accuracy. Figure 3 shows PSI-KT requires the least retraining time, retains the best prediction accuracy, and thus achieves the most favorable cost-accuracy trade-off (details in Appendix A.5.3).

## 4.3 INTERPRETABILITY OF REPRESENTATIONS

We now evaluate the interpretability of both learner-specific cognitive traits $s^\ell$ and the prerequisite graphs $\mathcal{A}$. We first show that our model captures learner-specific and disentangled traits that correlate with behavior patterns. Next, we show that our inferred graphs best align with ground truth graphs, and the edge weights predict causal support on downstream KCs.

### 4.3.1 LEARNER-SPECIFIC COGNITIVE TRAITS

For each learner, PSI-KT infers four latent traits, each with a clear dynamical role specified by the OU process (Eqs. 5-6). In contrast, high-performance baselines (AKT, DKT, and DKTF) describe learners via 16-dimensional embeddings solely constrained by network architecture and loss minimization. Another model QIKT constructs 3-dimensional embeddings with each element connected

---

[2] https://sites.google.com/site/assistmentsdata

[3] https://pslcdatashop.web.cmu.edu/DatasetInfo?datasetId=1198

Table 2: Prediction accuracy. FT indicates additional fine-tuning and ↑ indicates larger values are better. The **best model performance** is in bold and the 2nd best is underlined.

| Dataset | Experiment | HLR | PPE | DKT | DKTF | HKT | AKT | GKT | QIKT | PSI-KT |
|---|---|---|---|---|---|---|---|---|---|---|
| Assist12 | Within ↑ | $.54_{.03}$ | $.65_{.01}$ | $.65_{.03}$ | $.60_{.01}$ | $.55_{.01}$ | $\underline{.67}_{.02}$ | $.63_{.03}$ | $.63_{.03}$ | $\mathbf{.68}_{.02}$ |
| | Between ↑ | $.50_{.03}$ | $.50_{.02}$ | $.55_{.02}$ | $.51_{.01}$ | $.54_{.00}$ | $.58_{.02}$ | $\mathbf{.61}_{.02}$ | $\underline{.60}_{.02}$ | $\mathbf{.61}_{.03}$ |
| | w/ FT ↑ | $.52_{.02}$ | $.53_{.01}$ | $.58_{.00}$ | $.55_{.01}$ | $.55_{.00}$ | $\underline{.61}_{.00}$ | $\mathbf{.62}_{.02}$ | $.60_{.03}$ | $.62_{.02}$ |
| Assist17 | Within | $.45_{.01}$ | $.53_{.02}$ | $.57_{.02}$ | $.53_{.03}$ | $.52_{.03}$ | $.56_{.02}$ | $.56_{.04}$ | $\underline{.58}_{.02}$ | $\mathbf{.63}_{.02}$ |
| | Between | $.33_{.03}$ | $\underline{.51}_{.02}$ | $.51_{.00}$ | $.48_{.00}$ | $\underline{.51}_{.02}$ | $.47_{.01}$ | $\mathbf{.53}_{.02}$ | $.50_{.02}$ | $\mathbf{.53}_{.02}$ |
| | w/ FT | $.41_{.04}$ | $.51_{.00}$ | $.51_{.03}$ | $.53_{.01}$ | $.51_{.03}$ | $.51_{.02}$ | $\underline{.54}_{.03}$ | $.51_{.04}$ | $\mathbf{.56}_{.02}$ |
| Junyi15 | Within | $.55_{.02}$ | $.66_{.03}$ | $.79_{.03}$ | $.78_{.01}$ | $.63_{.02}$ | $\underline{.81}_{.02}$ | $.78_{.02}$ | $\underline{.81}_{.02}$ | $\mathbf{.83}_{.02}$ |
| | Between | $.48_{.02}$ | $.55_{.02}$ | $.76_{.00}$ | $.76_{.02}$ | $.61_{.01}$ | $.73_{.01}$ | $\underline{.77}_{.03}$ | $.76_{.03}$ | $\mathbf{.79}_{.03}$ |
| | w/ FT | $.52_{.00}$ | $.65_{.03}$ | $.81_{.01}$ | $\mathbf{.84}_{.01}$ | $.64_{.03}$ | $\underline{.83}_{.00}$ | $.79_{.03}$ | $.80_{.03}$ | $.80_{.02}$ |

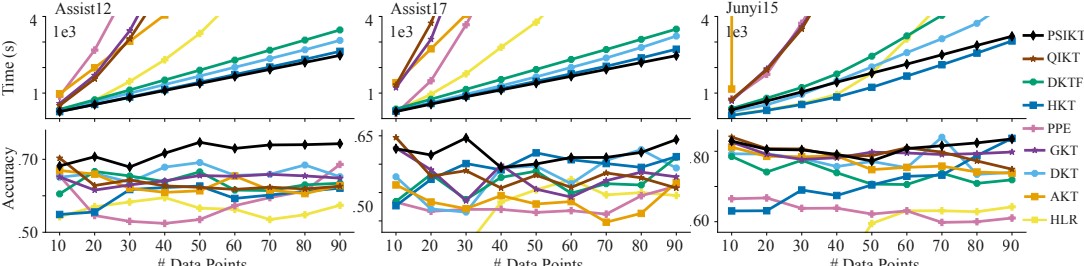

Figure 3: Continual learning. *(Top)* Cumulative training time. *(Bottom)* Prediction accuracy on the next 10 time steps. We omit results when time is above, or accuracy is below, the range of the axes.

to scores of knowledge acquisition, knowledge mastery, and problem-solving. We collectively refer to these learner-specific variables as *learner representations*. Here, we empirically show that PSI-KT representations provide superior interpretability. We ask that learner representations be 1) *specific* to individual learners, 2) *consistent* when trained on partial learning histories, 3) *disentangled* (i.e., component-wise meaningful, as in Bengio et al., 2013), and 4) and *operationally interpretable*, so that they can be used to personalize future curricula. We evaluate desiderata 1-3 with information-theoretic metrics (Table 3; see Appendix A.6 for details), and desideratum 4 with regressions against behavioral outcomes (Table 4).

**Specificity, consistency, and disentanglement.**
Learner representations $s$ should be maximally *specific* about learner identity $\ell$, which can be quantified by the mutual information $\mathrm{MI}(s; \ell) = \mathrm{H}(s) - \mathrm{H}(s \mid \ell)$ being high, where H denotes (conditional) entropy. Additionally, when we infer representations $s^{\ell_{\mathrm{sub}}}$ from different subsets of the interactions of a fixed learner, they should be *consistent*, i.e., each $s^{\ell_{\mathrm{sub}}}$ should be minimally informative about the chosen subset (averaged across subsets), such that $\mathbb{E}_{\ell_{\mathrm{sub}}}\mathrm{MI}(s^{\ell}; \ell_{\mathrm{sub}}) = \mathbb{E}_{\ell_{\mathrm{sub}}}[\mathrm{H}(s|\ell) - \mathrm{H}(s|\ell_{\mathrm{sub}})]$ should be low. Note that sequential subsets are unsuitable for this evaluation, since representations evolve in time to track learners' progression. Instead, we define subsets as groups of KCs

Table 3: Specificity, consistency, and disentanglement vs. best baseline.

| Metric | Dataset | Baseline | PSI-KT |
|---|---|---|---|
| Specificity $\mathrm{MI}(s; \ell)$ ↑ | Assist12 | $\mathbf{8.8}$ | $\underline{8.4}$ |
| | Assist17 | $\mathbf{10.1}$ | $\underline{10.0}$ |
| | Junyi15 | $\underline{13.5}$ | $\mathbf{14.4}$ |
| Consistency$^{-1}$ $\mathbb{E}_{\ell_{\mathrm{sub}}}\mathrm{MI}(s^{\ell}; \ell_{\mathrm{sub}})$ ↓ | Assist12 | $\underline{12.2}$ | $\mathbf{7.4}$ |
| | Assist17 | $\mathbf{6.4}$ | $6.4$ |
| | Junyi15 | $\underline{7.7}$ | $\mathbf{5.0}$ |
| Disentanglement $D_{\mathrm{KL}}(s\|\ell)$ ↑ | Assist12 | $\underline{2.3}$ | $\mathbf{7.4}$ |
| | Assist17 | $\underline{0.6}$ | $\mathbf{8.4}$ |
| | Junyi15 | $\underline{5.0}$ | $\mathbf{11.5}$ |

whose average presentation time is approximately uniform over the duration of the experiment (see Appendix A.6.1 for details). Lastly, learner representations should be *disentangled*, such that each dimension is individually informative about learner identity. We measure disentanglement with $D_{\mathrm{KL}}(s\|\ell) := \mathrm{H}(s) - \mathrm{H}(s \mid \ell)_{\mathrm{diag}}$, a form of specificity that ignores correlations across $s^{\ell}$ dimensions by estimating the conditional entropy only with diagonal covariances.

In empirical evaluations (Table 3), PSI-KT's representations offer competitive specificity despite being lower-dimensional, and outperform all baselines in consistency and disentanglement. While disentanglement aids interpretability (Freiesleben et al., 2022), it does not itself entail domain-specific meaning for representational dimensions. We now demonstrate that PSI-KT representations correspond to clear behavioral patterns, which is crucial for future applications in educational settings.

**Operational interpretability.** Having shown that PSI-KT captures specific, consistent, and disentangled learner features, we now investigate whether these features relate to meaningful aspects of future

behavior, which would be useful for scheduling operations for ITS. We indeed find that the learner representations of PSI-KT forecast interpretable behavioral outcomes, such as performance decay or initial performance on novel KCs. Concretely, consider the observed *one-step performance difference* $\Delta y_n^\ell := y_n^\ell - y_{n-1}^\ell$. We expect it to be lower for longer intervals $\tau_n^\ell = t_n^\ell - t_{n-1}^\ell$ due to forgetting. However, we recognize no clear trend when plotting $\Delta y_n^\ell$ over $\tau_n^\ell$ for the Junyi15 dataset (Fig. 4, top right). We can explain this observation because different learners forget on different time scales. Plotting the same test data instead over scaled intervals $\tau_n^\ell \alpha_n^\ell$ (Fig. 4, top center) shows a clear trend against an exponential fit (solid line) with less variability, demonstrating that $\alpha_n^\ell$ (derived from past data only) adjusts for individual learner characteristics and can be interpreted as a personalized rate of forgetting. Here, the choice of the factor $\alpha_n^\ell$ is motivated by our inductive bias (Eq. 4). The trend is much less clear for all baselines: Fig. 4 (top left) uses the best fitting component across all learner representations from all baselines (full results in Fig. 8 in Appendix A.6.4). Analogously, when we consider *initial performance on a novel KC*, we find for PSI-KT that $\tilde{\mu}_n^{\ell,k}$ (which aggregates mastery of prerequisites for KC $k$ at time $t_n$, see Eq. 5) explains it better than the best baseline Fig. 4 (bottom panels). Table 4 shows that these superior interpretability results are significant and hold across all datasets. In Appendix A.6.4, we discuss two more behavioral signatures (performance variability and prerequisite influence) and show they correspond to the remaining components $\gamma_n^\ell$ and $\sigma_n^\ell$.

### 4.3.2 PREREQUISITE GRAPH

PSI-KT infers a prerequisite graph based on all learners' data, which helps it to generalize to unseen learners. Beyond helping prediction, reliable prerequisite relations are an essential input for curriculum design, motivating our interest in their interpretability. Figure 5a shows an exemplary inferred subgraph with the prerequisites of a single KC. To quantitatively evaluate the graph, we (i) measure the alignment of the inferred vs. ground-truth graphs and (ii) correlate inferred prerequisite probability with a Bayesian measure of causal support obtained from unseen behavioral data.

**Alignment with ground-truth graphs.** We analyze the Junyi15 dataset, which uniquely provides human-annotated evaluations of prerequisite and similarity relations between KCs. We discuss here the alignment of prerequisites and leave similarity for Appendix A.7. The Junyi15 dataset provides both an expert-identified prerequisite for each KC, and crowd-sourced ratings (6.6 ratings on average on a 1-9 scale). To compare with expert annotations, we compute the rank of each expert-identified prerequisite relation $i \to k$ in the relevant sorted list of inferred probabilities $\{a^{jk}\}_{j=1}^K$ and take the harmonic average (mean reciprocal rank, MRR; Yang et al., 2014). Next, we compute the negative log-likelihood (nLL) of inferred edges $a^{ik}$ using a Gaussian estimate of the (rescaled) crowd-sourced ratings for the $i \to k$ KC pair. We finally calculate the Jaccard similarity (JS) between the set of inferred edges ($a^{ik} > 0.5$) and those identified by experts as well as crowd-sourced edges with average ratings above 5. The results in Table 5 (left columns) consistently highlight PSI-KT's superior performance across all criteria (see Appendix A.7.1 for details).

**Causal support across consecutive interactions.** For education applications, we are interested in how KC dependencies impact learning effectiveness. If KC $i$ is a prerequisite of KC $k$, mastering KC $i$ contributes to mastering KC $k$, indicating a causal connection. In this analysis, we show that inferred edge probabilities $a^{ij}$ (Eq. 7) correspond to causal support$_{i \to k}$ (Eq. 11), derived from behavioral data through Bayesian causal induction (Griffiths & Tenenbaum, 2009). Specifically, we model the relationship between a candidate cause $C$ and effect $E$, i.e., a pair of KCs in our case, while accounting for a constant background cause $B$, representing the learner's overall ability and the influences of other KCs. We consider two hypothetical causal graphs, where Graph 0 $G_{i \not\to k}$ represents

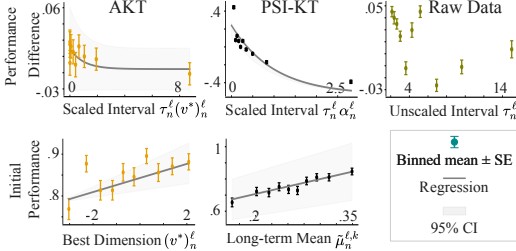

| Behavioural signature | Dataset | Best Baseline | PSI-KT |
|---|---|---|---|
| Performance difference | Assist12 | 0.01, .67 | **0.30**, <.001 |
| | Assist17 | −0.03, .30 | **0.56**, <.001 |
| | Junyi15 | 0.03, .06 | **0.72**, <.001 |
| Initial performance | Assist12 | 0.04, .01 | **0.54**, <.001 |
| | Assist17 | 0.05, .01 | **3.70**, <.001 |
| | Junyi15 | 0.04, .02 | **0.92**, <.001 |

Figure 4: Operational interpretability of representations, Junyi15 dataset. See text for axes labels and Appendix A.6.4 for additional results.

Table 4: Coefficients and $p$-values of regressions relating $\exp(-\alpha_n^\ell \tau_n^\ell)$ and $\tilde{\mu}_n^{\ell,k}$ to unseen behavioral data across datasets.

Table 5: *(Left)* Alignment of inferred graphs with annotated graphs for the Junyi15 dataset. *(Right)* Regression coefficients and $p$-values relating causal support to inferred edge probabilities. All baseline models either lack significance or negatively predict causal support (Appendix Fig. 12).

| Metric | MRR ↑ | JS expert ↑ | JS crowd ↑ | nLL ↓ | coefficient ↑, $p$-value ↓ | | |
|---|---|---|---|---|---|---|---|
| Dataset | | Junyi15 | | | Assist12 | Assist17 | Junyi15 |
| Best Baseline | .0082 | .0015 | .0047 | **3.03** | 1.05, .253 | 0.22, .792 | 0.42, .593 |
| PSI-KT | **.0086** | **.0019** | **.0095** | 4.11 | **1.15**, .003 | **0.28**, <.001 | **0.97**, <.001 |

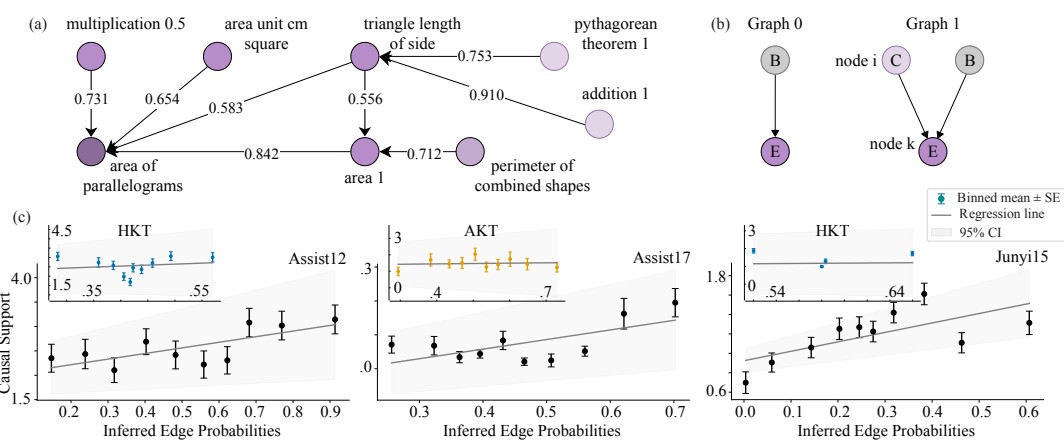

Figure 5: Graph interpretability. *(a)* Subgraph inferred by PSI-KT on the Junyi15 dataset, showing prerequisites of target KC 'area of parallelograms'. *(b)* Hypothesized causal graphs, where Graph 1 assumes a causal relationship exists from KC $i$ to KC $k$, while Graph 0 is the null hypothesis. *(c)* Regression of edge probabilities against causal supports. Insets show the best baseline model.

the null hypothesis of no causal relationship, and Graph 1 $G_{i \to k}$ assumes the causal relationship exists, i.e. correct performance on KC $i$ causally supports correct performance on KC $k$ (Fig. 5b). We estimate causal support for each pair of KCs $i \to k$ based on all consecutive interactions in the behavioral data $\mathcal{H}$ from KC $i$ at time $t_n$ to KC $k$ at time $t_{n+1}$, as a function of the difference in log-likelihoods of the two causal graphs (see Appendix A.7.3 for details):

$$\text{support}_{i \to k} := \log P(\mathcal{H} \mid G_{i \to k}) - \log P(\mathcal{H} \mid G_{i \not\to k}). \tag{11}$$

We then use regression to predict $\text{support}_{i \to k}$ as a function of edge probabilities $a^{ij}$ inferred from different models. The results are visualized in Figure 5c and summarized in Table 5 (right). The larger coefficients indicate that our inferred graphs possess superior operational interpretability (Sec. 4.3).

## 5  DISCUSSION

We propose PSI-KT as a novel approach to knowledge tracing (KT) with compelling properties for intelligent tutoring systems: superior **p**redictive accuracy, excellent continual-learning **s**calability, and **i**nterpretable representations of learner traits and prerequisite relationships. We further find that PSI-KT has remarkable predictive performance when trained on small cohorts whereas baselines require training data from at least 60k learners to reach similar performance. An open question for future KT research is how to combine PSI-KT's unique continual learning and interpretability properties with performance that grows beyond this extreme regime. We use an analytically marginalizable Ornstein-Uhlenbeck process for knowledge states in PSI-KT, resulting in an exponential forgetting law, similar to most recent KT literature. Future work should support ongoing debates in cognition by offering alternative modeling choices for memory decay (e.g., power-law; Wixted & Ebbesen, 1997), thus facilitating empirical studies at scale. And while our model already normalizes reciprocal dependencies in the prerequisite graph, we anticipate that enforcing regional or global structural constraints, such as acyclicity, may benefit inference and interpretability. Although we designed PSI-KT with general structured domains in mind, our empirical evaluations were limited to mathematics learning by dataset availability. We highlight the need for more diverse datasets for structured KT research to strengthen representativeness in ecologically valid contexts. Overall, our work combines machine learning techniques with insights from cognitive science to derive a predictive and scalable model with psychologically interpretable representations, thus laying the foundations for personalized and adaptive tutoring systems.

ACKNOWLEDGMENTS

The authors thank Nathanael Bosch and Tim Xiao for their helpful discussion, and Seth Axen for code review. The authors thank the International Max Planck Research School for Intelligent Systems (IMPRS-IS) for supporting Hanqi Zhou. This research was supported as part of the LEAD Graduate School & Research Network, which is funded by the Ministry of Science, Research and the Arts of the state of BadenWürttemberg within the framework of the sustainability funding for the projects of the Excellence Initiative II. Funded by the Deutsche Forschungsgemeinschaft (DFG, German Research Foundation) under Germany's Excellence Strategy – EXC number 2064/1 – Project number 390727645. CMW is supported by the German Federal Ministry of Education and Research (BMBF): Tübingen AI Center, FKZ: 01IS18039A.

ETHICS STATEMENT

We evaluated our PSI-KT model on three public datasets from human learners, which all anonymize the data to protect the identities of individual learners. Although PSI-KT aims to improve personalized learning experiences, it infers cognitive traits from behavioral data instead of using learners' demographic characteristics (e.g., age, gender, and the name of schools provided in the Assistment 17 dataset), to avoid reinforcing existing disparities.

Evaluations of structured knowledge tracing in our paper are limited by dataset availability to precollege mathematics. To ensure a broader and more ecologically valid assessment, it is essential to explore diverse datasets across various domains (e.g., biology, chemistry, linguistics) and educational stages (from primary to college level). This will allow for a more comprehensive understanding of the role of structure in learning.

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
