# OpenReview forum: "Predictive, scalable and interpretable knowledge tracing on structured domains"
_ICLR.cc/2024/Conference — ICLR 2024 spotlight_

### Official Review · Reviewer_da66 · 2023-10-25

**Soundness:** 2 fair
**Presentation:** 2 fair
**Contribution:** 3 good
**Rating:** 6
**Confidence:** 4

**Summary:**

This paper constructs a scientifically sound model for the knowledge tracing problem that takes into account past performance, prerequsite knowledge graphs, and individual learner traits. They compare this to a number of other methods for predicting learner performance using public data and exceed the baseline.

**Strengths:**

Predictive Accuracy was reasonable and well evaluated. The data used in the experiments was relevant and allowed reasonable evaluation. The graphs and tables produced were helpful in following the results.

In terms of the 4 primary dimensions used for an ICLR Review

- Originality: Combining knowledge tracing and knowledge mapping into one method is a nice combination of ideas into one framework.

- Quality: Quality was good, useful data of a reasonable size with a good baseline of comparison to other methods. In terms of basic accuracy this was well presented.

- Clarity: The presentation overall left a lot to be desired in this paper but the graphs and tables were

- Significance: The primary significance of these results is in the interpretability of the results.

**Weaknesses:**

Most of the focus of this paper was on the accuracy. Interpretability and scalability were not well evaluated and much of that was in the form of "correct by construction".

The prerequisite graph was interesting, although the correctness of the graph was not well quantified.

And although I thought the accuracy beat the provided baseline and had sufficient data to support that, I do not think the results are good, only that they are better than the baseline. For a binary problem, getting accuracy of 55-80 is not a strong result.

**Questions:**

I would also like to more details on the datasets, particularly from the perspective of diversity. Claims about educational effectiveness and knowledge graphs that do no reflect a sufficient cross section are suspect at best and can be actively harmful.

---

> ### Author Response · Authors · 2023-11-18
> **Response to Reviewer da66**
>
> We thank the reviewer for their feedback and address their questions below.
> > accuracy of 55-80 is not a strong result
>
> We believe there is a misunderstanding. Table 2 reports predictive accuracies on several tasks, and we only obtain accuracies around 55% for between-learner generalization on Assist17. This task is deliberately designed to be difficult since the lack of historical data prevents learner personalization (note that baselines perform even worse). Accuracies for within-learner predictions (most commonly reported in related work), are considerably higher (63% to 83%), outperforming all baselines (incl. GKT and QIKT added based on reviewer feedback).
>
> We realize it may not be obvious why higher predictive accuracies are not common in KT, which is known to be difficult due to selection bias: data are collected from learning systems designed to engage learners with KCs that are at the edge of their abilities. Thus, the data collection process inadvertently focuses on learner/KC-pairs where performance is particularly hard to predict.
>
> > more details on the datasets, particularly from the perspective of diversity. Claims about educational effectiveness and knowledge graphs that do no reflect a sufficient cross section are suspect at best and can be actively harmful.
>
> We appreciate these important concerns. We address them using more cautious language in the introduction, and more details on our datasets and the selection criteria in the Appendix.
>
> While we expect our model to enhance KT in structured domains beyond mathematics, we also want to avoid implying, even indirectly, that our model is universally applicable. Accordingly we propose the following edits to the introduction (~~removed~~, *added*):
> “[…] we present PSI-KT, a novel approach for inferring interpretable learner-specific cognitive traits and a shared ~~knowledge~~ graph of *concept* prerequisites ~~relationships~~. We demonstrate ~~the effectiveness of~~ our approach on three real-world educational datasets *covering structured domains*,  where our model outperforms existing baselines […]”.
>
> We will clarify our choice of datasets in A.3.2. In brief, we require datasets in (1) structured domains with (2) KC labels and (3) high temporal resolution. These conditions rule out Statics2011 (2), Assist09 and Assist15 (3) and Junyi20 (3).
>
> > - Strengths: The primary significance of these results is in the interpretability of the results.
> > - Interpretability and scalability were not well evaluated and much of that was in the form of "correct by construction".
>
> We are glad that the reviewer acknowledges the significance of the interpretability. A strength of our generative model is that we can explicitly define interpretable dependencies. However, far from assuming the interpretability is “correct by construction”, we spend the entire Sec. 4.3 to critically test this hypothesis using multiple metrics (specificity, consistency, disentanglement, and operational interpretability). Here, the first three are information theoretical metrics, which are agnostic to any preconceived notion of interpretability precisely to avoid the risk of assuming “correct by construction”. We also analyze the interpretability of the inferred graph (see next question).
>
> While we appreciate suggestions for further interpretability analysis, we believe the interpretability of our model is very carefully evaluated and is one of our most significant results (as acknowledged by the reviewer). Given the diverse interpretability metrics that were carefully chosen to avoid unfairly favoring our model, we find it remarkable that our model is consistently superior than baselines (Tables 3-5).
>
> We apologize for any confusion regarding scalability. In educational contexts, we specifically care about scalability where data from each learner increases over time, and aim to maintain efficient retraining and robust prediction accuracy as new data emerges. Figure 3 shows our model excels in both aspects, making it a compelling choice in future educational settings.
>
> > The prerequisite graph was interesting, although the correctness of the graph was not well quantified.
>
> We are grateful that the reviewer is interested in our prerequisite graph. To evaluate inferred graphs (across now 7 models), we tested graph *alignment* against human-annotated prerequisites (both expert and crowd-sourced) with 3 metrics, and graph *operational interpretability* using Bayesian causal analysis to predict future learning outcomes from inferred edges, with favorable results (Table 5, and Appendix Fig. 12).
>
> Additionally, given suggestions from reviewers (J4qa) and (rAUs), we now support these quantifications with an ablation study that shows our inferred graph is a key ingredient for our model to achieve its superior predictive accuracy (Appendix Fig. 13 and Table 16).
> We remain open for additional analysis to improve the quantification of the correctness of the graph and welcome your suggestions.

---

> > ### Author Response · Authors · 2023-11-21
> > **Follow-up Response to Reviewer da66**
> >
> > We have now updated the revised manuscript and appendix to incorporate your suggestions. We would be grateful for feedback pointing to specific improvements, hope that by addressing your comments, you would consider updating your evaluation. We summarize the main changes below.
> >
> > Key updates:
> >
> > 1. Clarifications: We have addressed all the queries in our revised manuscript and extended reporting of the datasets in appendix. Particularly, we clearly point out diversity limitations in the discussion section:
> > “Although we designed PSI-KT with general structured domains in mind, our empirical evaluations were limited by dataset availability to mathematics learning. We highlight the need for more diverse datasets for structured KT research to strengthen its representativeness in ecologically valid contexts.
> > 2. Addition of Additional Baselines: Based on suggestions from reviewers *k3Pn* and *J4qa*, we have included new baseline comparisons (GKT and QIKT in Figs 2-3, 12 and Tables 2, 9-13, 15), providing a more comprehensive evaluation of our approach. We clarified in our reply above why KT problems have different accuracy ranges than other binary classification problems, and this expanded set of baselines provides additional support for the benefits of our model.
> > 3. Ablation Study: To further clarify the contributions of the different components of our model, we have conducted an ablation study.  These results are now included in A.8, which show crucial contributions from all three components.
> >
> > We are hopeful that these enhancements address the concerns raised by the reviewer.

---

> ### Comment · Reviewer_da66 · 2023-11-21
>
> Thank you, authors, for all the clarifications, updates, and changes. I still think that the strongest parts of this paper are in the interpretability of the work and the knowledge graph, but after clarification. I have updated my review based on your revisions and I hope to see this work and any follow on papers in publication soon.

---

### Official Review · Reviewer_k3Pn · 2023-10-27

**Soundness:** 3 good
**Presentation:** 2 fair
**Contribution:** 2 fair
**Rating:** 5
**Confidence:** 4

**Summary:**

This paper introduces PSI-KT, a generative knowledge tracing method that places emphasis on predictive accuracy, scalable inference, and interpretability. PSI-KT models both the cognitive processes of students and the underlying knowledge prerequisite structure. Extensive experimental results clearly showcase the method's superiority over various baselines from multiple angles.

**Strengths:**

1. The proposed method is designed carefully and comprehensive, focusing on mutiple perspectives of the knowledge tracing task.
2. The motivation is meaningful that this paper focus on the pain point about interpretability of the knowledge tracing field.
3. The paper is well-structured.

**Weaknesses:**

1. The method's description is not sufficiently clear. As indicated in the appendix, PSI-KT also employs neural networks to generate cognitive parameters. However, the main body of the paper only briefly touches upon this aspect, potentially leading to the misconception that PSI-KT is not a deep learning approach.
2. The experimental setup lacks persuasiveness. As demonstrated in Table 1, two datasets contain over 10,000 learners, yet the authors chose to use only 100-1,000 learners as training data. Conducting experiments with a small dataset may unfairly disadvantage deep learning baselines, which can effectively leverage the abundance of available data. The reasoning provided, "to simulate real-world data constraints in education," may not hold in the context of the vast amount of student learning data generated today.
3. The introduction of interpretable KT methods is not comprehensive. For instance, recent approaches like IKT, ICKT, and QIKT [1, 2, 3] incorporate interpretable psychological and cognitive modules into their methods. These relevant methods are not referenced in this paper, let alone included as baselines in the experiments.
4. The assessment of the model's interpretability is not entirely convincing. The limited dimensionality of hidden learner representations in deep learning methods (e.g., DKT, AKT) at just 16 may constrain the neural networks' capabilities. Furthermore, there is no supporting evidence indicating that the learner representations of PSI-KT and these deep learning baselines capture the same underlying student features, making direct comparisons less rational.
5. Perhaps conducting case studies of PSI-KT could offer a more intuitive understanding of its interpretability, such as visualizing trends in students' knowledge mastery, as shown in Figure 1(a).

**Questions:**

1. Why did the authors choose to experiment with only a limited portion of the datasets? The explanation provided, "to simulate real-world data constraints in education," may benefit from additional clarification.
2. Could the authors consider using more recent interpretable deep learning methods like QIKT as their baseline comparisons? Doing so could enhance the credibility of the study.
3. Is there a specific reason why the authors did not provide case studies to visually demonstrate the model's interpretability, as has been done in previous KT research?
4. Could the authors elaborate on the detailed rationale behind using mutual information between PSI-KT's learned parameters and the hidden vectors of baselines to measure interpretability? Further explanation would enhance the understanding of the experiments.

---

> ### Author Response · Authors · 2023-11-20
> **Response to Reviewer k3Pn - Part 1/3**
>
> We thank the reviewer for their valuable feedback, which has allowed us to improve the manuscript by including additional experiments with new baselines and more data, and also adding clarifications on methodology, interpretability, and case studies.
>
> ### Q1: Experiments on more learners
> > Why did the authors choose to experiment with only a limited portion of the datasets? The explanation provided, "to simulate real-world data constraints in education," may benefit from additional clarification.
>
> We thank the reviewer for spotting that “real-world data constraints” is imprecise and leads to confusion. We now (a) clarify our focus on small-to-moderate cohort sizes and (b) extend performance evaluations to all learners. In detail:
>
> 1. We reformulated the start of Section 4.1:
> “In our evaluations, we mainly focus on prediction and generalization when training on 10 interactions from up to 1000 learners. Good KT performance with little data is key in practical ITS to minimize the number of learners on an experimental treatment (principle of equipoise, similar to medical research [1]), to mitigate the cold-start problem, and extend the usefulness of the model to classroom-size groups. To provide ITS with a basis for adaptive guidance and long-term learner assessment, we always predict the 10 next interactions.”
> 2. We have run additional experiments and now report performance for the maximum number of learners available in each dataset after preprocessing (Figure 2, Table 11).  We discuss them in Section 4.1:
> “Figure 2 shows that PSI-KT's *within-learner prediction* performance is robustly above baselines for all but the largest cohort size (>60k learners, Junyi15), where all deep learning models perform similarly.
> The *between-learner generalization* accuracy of these models when tested on 100 out-of-sample learners is shown in Table 2, where fine tuning indicates that parameters were updated using (10-point) learning histories from the unseen learners. PSI-KTshows overall superior generalization except on Junyi15 (when fine-tuning).”
>
> We are grateful for the opportunity to clarify our focus on the < 1000 learners setting, and for encouraging us to look beyond it into large cohorts. The differences in behavior between models warrant further research, which we have reflected in the discussion:
> “We further find that PSI-KT has remarkable predictive performance when trained on small cohorts whereas baselines require training data from at least 60k learners to reach similar performance. An open question for future KT research is how to combine PSI-KT's unique continual learning and interpretability properties with performance that grows beyond this extreme regime.”
>
>
> ### Q2: QIKT baseline
> > The introduction of interpretable KT methods is not comprehensive. For instance, recent approaches like IKT, ICKT, and QIKT incorporate interpretable psychological and cognitive modules into their methods.
> > Could the authors consider using more recent interpretable deep learning methods like QIKT as their baseline comparisons? Doing so could enhance the credibility of the study.
>
> We thank the reviewer for the valuable feedback and pointing us towards additional baselines. We found that QIKT is an excellent baseline to add (together with GKT [5], proposed by reviewer *J4qa*) and are pleased that our more comprehensive assessment (Figure 2-3, 12 and Table 2, 9-13, 15) provides even stronger support for our contributions.
>
> We have improved the related work (Sec. 2), now covering interpretable KT approaches IKT [2], IEKT [3] and QIKT [4].
>
> Adding even more baselines might still be valuable, but we were unable to include more because (1) the official IKT code lacks a key component (tree structure) making a fair comparison challenging; (2) while we were unfortunately unable to find a reference for ICKT, we found the similarly named IEKT and now reference it in the related work. We did not add it to evaluations as we believe QIKT provides a superior baseline. Should ICKT differ from IEKT, we welcome a reference for potential future inclusion.

---

> > ### Author Response · Authors · 2023-11-20
> > **Response to Reviewer k3Pn - Part 2/3**
> >
> > ### Q3: Interpretability explanations
> > > The assessment of the model's interpretability is not entirely convincing. The limited dimensionality of hidden learner representations in deep learning methods (e.g., DKT, AKT) at just 16 may constrain the neural networks' capabilities.
> >
> > We agree that low dimensional representations can constrain model performance. However, note that our model has even lower 4-dimensional learner representations. Further, low dimensions may constrain predictive performance (of the entire model) but not the interpretability (of individual components of the learner representation).
> >
> > > - no supporting evidence indicating that the learner representations of PSI-KT and these deep learning baselines capture the same underlying student features, making direct comparisons less rational.
> > > - Could the authors elaborate on the detailed rationale behind using mutual information
> >
> > Thank you for bringing up these two points, which are indeed related: the rationale for expressing specificity, consistency, and disentanglement in terms of mutual information (Sec. 4.3.1) is precisely to avoid applying our preconceived notions of interpretable cognitive traits to the evaluation of baseline models. As an information-theoretical quantity, mutual information only quantifies the *amount* of information that, e.g., a learner representation contains about the learner’s identity (for the specificity metric). It is agnostic to the form in which this information is encoded in the learner representation. If a baseline model finds different underlying learner features that do not match our psychologically-motivated cognitive traits then the model can still have a high specificity score as long as the identified learner features contain information to identify the learner in some way. The same holds for consistency and disentanglement. We will add a shortened form of this clarification to the paper.
> >
> > ### Q4: Clarifications on neural networks
> > > The method's description is not sufficiently clear. As indicated in the appendix, PSI-KT also employs neural networks to generate cognitive parameters.
> >
> > We are very grateful for this feedback and have improved the clarity of the methods by directly indicating our use of a neural network in the title of Sec 3.2 (“Approximate Bayesian Inference and Amortization with a Neural Network”) and introducing the terminology “inference network” in the first sentence:
> >
> > Before: “We now describe how we learn the generative model parameters $\theta$ and infer the latent states $s, z$ introduced in Section 3.1.”
> >
> > After: “[…] and how we infer the latent states $s, z$ introduced in Section 3.1 using a neural network (``inference network'').”
> >
> > ### Q5: Case study visualization
> > > Perhaps conducting case studies of PSI-KT could offer a more intuitive understanding of its interpretability, such as visualizing trends in students' knowledge mastery, as shown in Figure 1(a).
> >
> > Thank you for this suggestion. We have added a visual example (Fig. 11 in A.6.5) of how latent knowledge states evolve in response to learning interactions, similar to our schematic Figure 1. We note that our model can estimate knowledge states at all times and not just interaction times, which allows us to use natural time in the x-axis and display knowledge states with curves instead of using the discrete color maps common in the KT literature.
> >
> > In light of these clarifications, we would appreciate if you considered increasing our score to reflect our revisions. If not, could you let us know any additional changes you would like to see in order for this work to be accepted?

---

> > > ### Author Response · Authors · 2023-11-20
> > > **Response to Reviewer k3Pn - Part 3/3**
> > >
> > > References:
> > >
> > > [1] Burkholder, L. (2021). Equipoise and ethics in educational research. Theory and Research in Education, 19(1), 65-77.
> > >
> > > [2] Minn, S., Vie, J. J., Takeuchi, K., Kashima, H., & Zhu, F. (2022, June). Interpretable knowledge tracing: Simple and efficient student modeling with causal relations. In Proceedings of the AAAI Conference on Artificial Intelligence (Vol. 36, No. 11, pp. 12810-12818).
> > >
> > > [3] Long, T., Liu, Y., Shen, J., Zhang, W., & Yu, Y. (2021, July). Tracing knowledge state with individual cognition and acquisition estimation. In Proceedings of the 44th International ACM SIGIR Conference on Research and Development in Information Retrieval (pp. 173-182).
> > >
> > > [4] Chen, J., Liu, Z., Huang, S., Liu, Q., & Luo, W. (2023). Improving interpretability of deep sequential knowledge tracing models with question-centric cognitive representations. arXiv preprint arXiv:2302.06885.
> > >
> > > [5] Nakagawa, H., Iwasawa, Y., & Matsuo, Y. (2019, October). Graph-based knowledge tracing: modeling student proficiency using graph neural network. In IEEE/WIC/ACM International Conference on Web Intelligence (pp. 156-163).

---

### Official Review · Reviewer_J4qa · 2023-10-29

**Soundness:** 3 good
**Presentation:** 3 good
**Contribution:** 3 good
**Rating:** 8
**Confidence:** 3

**Summary:**

This paper proposes a probabilistic state-space generative approach named PSI-KT by explicitly modeling individual cognitive traits and shared knowledge graph of prerequisite relationships to achieve predictive, scalable and interpretable knowledge tracing, inspired by cognitive science and pedagogical psychology. The author conducts extensive experiments on three datasets to demonstrate that PSI-KT can achieve superior predictive accuracy, scalable inference in continual-learning settings, and interpretability of learners’ cognitive traits and prerequisite graphs. The paper’s contributions are as follows:

1.The paper proposes a novel hierarchical probabilistic state-space model for knowledge tracing by introducing individual cognitive traits and prerequisite shared knowledge graph.

2.Unlike recent discriminative KT models that utilize cross-entropy loss, PKI-KT distinguishes itself by introducing a psychologically-inspired probabilistic generative model, which leverages approximate Bayesian inference and variational continual learning techniques for model optimization.

3.Extensive experiments demonstrate that PKI-KT achieves impressive results in multi-step predictive accuracy and scalable inference in continual-learning settings. Moreover, novel confirmatory experiments further validate the specificity, consistency, disentanglement, and operational interpretability of individual cognitive traits, as well as the reliability of the inferred prerequisite graph.

**Strengths:**

1.Good textual expression, mathematical notation, and formula derivations. The paper provides a clear description of motivation, problem definition, and experimental setup, along with professionally presented mathematical expressions.

2.The motivation is both novel and reasonable. PSI-KT takes into account students’ individual cognitive traits and the prerequisite knowledge graphs while modeling students' knowledge states.

3.The proposed method is intriguing. PSI-KT applies a Probabilistic State-Space Model to model students' knowledge states in KT. It introduces a three-level hierarchical structure, utilizes approximate Bayesian inference for generating students' knowledge states and cognitive traits, and optimizes model parameters using the Evidence Lower Bound (ELBO) instead of the common cross-entropy used in recent discriminative KT models.

4.The paper includes extensive confirmatory experiments with detailed and favorable results. In addition to conducting rich experiments on predicting student performance in both within-learner and between learner settings, the authors also carries out numerous analytical validation experiments concerning the representation of cognitive traits and the inferred knowledge prerequisite relationships, all of which have yielded positive outcomes.

**Weaknesses:**

1.The cognitive traits in the paper lack somewhat interpretability. While the authors have conducted extensive validation experiments on the representation of cognitive traits, considering that the paper introduces cognitive traits from the perspectives of cognitive science and psychology, it is advisable to explicitly state in the text which specific cognitive psychology traits the four dimensions of cognitive traits represent. This would help readers better understand the meaning and significance of these traits.

2.Experiments are somewhat insufficient. Although the authors have conducted an extensive array of analytical and validation experiments, there is a notable absence of ablation study to demonstrate the effectiveness of the two proposed motivations in the paper, namely cognitive traits and the prerequisite relationship graph, on PSI-KT. Furthermore, given the mention of the use of the prerequisite graph in the paper, it seems somewhat inadequate not to include some explicit baseline models that utilize knowledge concept graphs for comparison.

**Questions:**

1.Could the authors provide some explanations about the four dimensions of cognitive traits and how they represent specific characteristics of students? It would be particularly helpful if these dimensions can be correlated with concepts from cognitive science. Additionally, I'm interested in an experimental analysis of the impact of the other two dimensions.

2.Have the authors considered supplementing with essential ablation study and adding baseline models that explicitly take into account the knowledge graph structure, such as GKT[1] or SKT[2], the latter of which also considers prerequisite relationship between concepts?

[1] Nakagawa, Hiromi, Yusuke Iwasawa, and Yutaka Matsuo. "Graph-based knowledge tracing: modeling student proficiency using graph neural network." IEEE/WIC/ACM International Conference on Web Intelligence. 2019.

[2] Tong, Shiwei, et al. "Structure-based knowledge tracing: An influence propagation view." 2020 IEEE international conference on data mining (ICDM). IEEE, 2020.

---

> ### Author Response · Authors · 2023-11-21
> **Response to Reviewer J4qa**
>
> We are grateful for the strong evaluation and detailed feedback on our analyses.
>
> > Could the authors provide some explanations about the four dimensions of cognitive traits and how they represent specific characteristics of students? It would be particularly helpful if these dimensions can be correlated with concepts from cognitive science. Additionally, I'm interested in an experimental analysis of the impact of the other two dimensions.
>
> We thank the reviewer for pointing out potential confusion with the interpretation of the cognitive traits. We have now revised the methods section to more clearly explain these dimensions (Sec. 3.1):
> “Specifically, $\alpha^\ell$ represents the forgetting rate [1,2]  $\mu^\ell$ (via $\tilde{\mu}^{\ell,n}_k$) captures long-term memory consolidation [3] for practiced KCs and expected performance for novel KCs, $\sigma^\ell$ is knowledge volatility, and $\gamma^\ell$ indicates transfer ability [4] from performance on prerequisite KCs.”
>
> Following the suggestion, we have performed additional interpretability experiments that relate the other two dimensions (transfer ability and knowledge volatility), to appropriate behavioral measures (Fig. 10 in Appendix A.6.4). In brief, we relate transfer ability to successful transitions on the prerequisite graph and knowledge volatility to temporal variability in individual performance. In all cases, our model achieves the highest match between parameters and behavior. We believe these additions have substantially improved the interpretability of the cognitive traits captured by our model.
>
>
> > Have the authors considered supplementing with essential ablation study and adding baseline models that explicitly take into account the knowledge graph structure, such as GKT[1] or SKT[2], the latter of which also considers prerequisite relationship between concepts?
>
> We are grateful for the suggestion of an ablation analysis (also mentioned by reviewer *rAUs*) and the recommendation of additional baselines. The suggestion of performing ablation analyses was incredibly helpful, and allowed us to discover that all components of the model are crucial for performance. Additionally, the addition of a strong structure-aware baseline (GKT) has strengthened the credibility and thoroughness of support for our model.
>
> In a new ablation analysis (Table 16 and Figure 13, Appendix A.8), we ablate the graph structure, the individual traits, and learner dynamics, observing significant drops in accuracy in each case. No single ablation reliably corresponds to the largest drop in accuracy across datasets. Thus, we believe that all components make important contributions to the robustness and generalizability of our model.
>
> We have also added GKT to our set of baselines (along with an additional model QIKT [5] mentioned by other reviewer *k3Pn*). GKT represents one of the strongest baselines and expands our comprehensive assessment of alternative models (Figure 2-3, 12 and Table 2, 9-13, 15), which altogether provides even stronger support for our model.
>
> While SKT has a number of innovative methods for modeling both similarity and prerequisite structures, we could not find how to perform data splitting to calculate the graph (either in the code base or paper). Specifically, the graph is calculated from performance statistics (i.e., correct/incorrect counts), which cannot be directly used to predict the same performance data (to avoid circularity). This same circularity is not an issue when the graph inference is part of an end-to-end learning pipeline, as in ours and other models. For this reason a fair comparison seems challenging.
>
> In sum, we have clarified the interpretability of cognitive traits, added additional analyses for the other two dimensions, performed an ablation analysis, and added an additional model to our baselines. We believe this has greatly strengthened the paper and welcome the reviewer to consider revising their score to reflect these changes.
>
> References:
>
> [1] Ebbinghaus, H. (1885). Über das gedächtnis: untersuchungen zur experimentellen psychologie. Duncker & Humblot.
>
> [2] Averell, L., & Heathcote, A. (2011). The form of the forgetting curve and the fate of memories. Journal of mathematical psychology, 55(1), 25-35.
>
> [3] Meeter, M., & Murre, J. M. (2004). Consolidation of long-term memory: evidence and alternatives. Psychological Bulletin, 130(6), 843.
>
> [4] Bassett, D. S., & Mattar, M. G. (2017). A network neuroscience of human learning: potential to inform quantitative theories of brain and behavior. Trends in cognitive sciences, 21(4), 250-264.
>
> [5] Chen, J., Liu, Z., Huang, S., Liu, Q., & Luo, W. (2023). Improving interpretability of deep sequential knowledge tracing models with question-centric cognitive representations. arXiv preprint arXiv:2302.06885.

---

### Official Review · Reviewer_rAUs · 2023-11-01

**Soundness:** 3 good
**Presentation:** 3 good
**Contribution:** 3 good
**Rating:** 8
**Confidence:** 3

**Summary:**

The paper presents PSI-KT, a novel knowledge-tracing model that combines individual learning dynamics with structural influences from prerequisite relationships. PSI-KT uses Bayesian inference to model learner-specific cognitive traits and shared prerequisite graphs. Evaluated on real educational datasets, PSI-KT achieves superior predictive accuracy and scalability while also providing interpretable representations of learners and knowledge structure. The model helps advance personalized intelligent tutoring systems by combining insights from cognitive science and machine learning. PSI-KT demonstrates how explicitly modeling psychological principles within AI systems can enhance performance and interpretability.

**Strengths:**

* The model is designed based on psychological principles and evaluated on multiple datasets. The experiments demonstrate predictive accuracy, scalability, and interpretability. The paper is technically strong in its probabilistic modeling and inference methodology.
* The paper is well-written and provides intuitive explanations of the model components. The background gives a clear overview of knowledge tracing and related work.
*The model advances knowledge tracing for intelligent tutoring systems by enhancing predictive accuracy, scalability, and interpretability. The interpretable representations of learners and knowledge structure provide an important basis for personalized education. The integration of cognitive science and AI is significant for developing systems that leverage psychological insights.

**Weaknesses:**

* The evaluations focus on three specific educational datasets. Testing on a more diverse range of datasets could better reveal the model's capabilities and limitations. The authors could discuss what other domains or data characteristics pose challenges.
* Long-term retention modeling could be enhanced. The current exponential decay may be simplistic. Exploring more complex forgetting functions based on memory research literature could improve long-term predictions.
* While superior overall, some accuracy metrics are comparable to certain baselines. Further ablation studies could provide insight into which model components contribute most to accuracy gains.

**Questions:**

* Could you provide insights into the dataset limitations and discuss potential challenges in applying the model to other educational domains or datasets?
* Have you considered exploring more complex forgetting functions based on memory research literature to improve long-term predictions?
* Could you perform ablation studies to dissect the contributions of different model components to predictive accuracy, providing insights into the model's strengths?

---

> ### Author Response · Authors · 2023-11-21
> **Response to Reviewer rAUs**
>
> We thank the reviewer for the valuable feedback. We address them below and also increase the manuscript by adding more clarifications and new ablation studies.
>
> > Could you provide insights into the dataset limitations and discuss potential challenges in applying the model to other educational domains or datasets?
>
> We thank the reviewer for the thorough feedback on our paper. We agree that the representativeness of empirical evaluations should be a central concern for ours and other models, and now highlight it in the discussion:
> ​​“Although we designed PSI-KT with general structured domains in mind, our empirical evaluations were limited to mathematics learning by dataset availability. We highlight the need for more diverse datasets for structured KT research to strengthen representativeness in ecologically valid contexts.”
>
> We expand on limitations in Appendix A.3.2 where we now describe in detail the datasets and the criteria we used to select them (also in response to reviewer *da66*). In brief, our choice of datasets was constrained by the following requirements: we require data sets in structured domains with (1) KC labels and (2) high temporal resolution. These conditions rule out Statics2011 (1), Assist09 and Assist15 (2), and Junyi20 (2).
>
> > - Long-term retention modeling could be enhanced. The current exponential decay may be simplistic. Exploring more complex forgetting functions based on memory research literature could improve long-term predictions.
> > - Have you considered exploring more complex forgetting functions based on memory research literature to improve long-term predictions?
>
> The reviewer is right that exponential forgetting alone, while common in the psychophysics [1] and recent KT literature [2], may be simplistic [3]. That is the reason why we extend the vanilla Ornstein-Uhlenbeck process with a reversion mean for long-term retention (now better clarified in Sec 3.1) [3], and parameterize the process with learner- and time-dependent cognitive traits, resulting in a very expressive model of memory dynamics.
>
> Despite our good empirical results with this extended OU model (see also new ablation results below), we do agree with the reviewer that supporting a variety of forgetting functions is key to bringing the benefits of memory research to KT (and conversely), a concern we now reflect in the discussion:
> “We use an analytically marginalizable OU process for knowledge states in PSI-KT, resulting in an exponential forgetting law, similar to most recent KT literature. Future work should support ongoing debates in cognition by offering alternative modeling choices for memory decay e.g., power-law [4], thus facilitating empirical studies at scale. “
>
> Because the scalability properties of stochastic process capturing alternative forgetting functions (e.g., power law) require additional investigation, we intend to address this in future work.
>
> > Could you perform ablation studies to dissect the contributions of different model components to predictive accuracy, providing insights into the model's strengths?
>
> We are very grateful to the reviewer for the suggestion of performing ablation studies, which was suggested by the reviewer *J4qa* as well. When we ablate any of the graph structure, the individual traits, and learner dynamics, we observe significant drops in accuracy across all three datasets. No single ablation reliably corresponds to the largest drop in accuracy across datasets. Thus, we believe that all components together allow our model to adapt to a variety of educational contexts and datasets, thereby enhancing its robustness and generalizability.
>
> We present the results and their interpretation in Appendix A.8 (Table 16 and Figure 13) and summarize them in Section 4.1:
> "The advantage of PSI-KT comes from its combined modeling of KC prerequisite relations and individual learner traits that evolve in time."
>
> Overall, we are encouraged that these insightful suggestions resulted in clarifications and analyses that strengthen the relevance of our contribution, helping us to reach out beyond the machine learning community. We hope that the reviewer will consider revising our score and we welcome further suggestions for improvement.
>
> References:
>
> [1] Ebbinghaus, H. (1885). Über das gedächtnis: untersuchungen zur experimentellen psychologie. Duncker & Humblot.
>
> [2] Nagatani, K., Zhang, Q., Sato, M., Chen, Y. Y., Chen, F., & Ohkuma, T. (2019, May). Augmenting knowledge tracing by considering forgetting behavior. In The world wide web conference (pp. 3101-3107).
>
> [3] Averell, L., & Heathcote, A. (2011). The form of the forgetting curve and the fate of memories. Journal of mathematical psychology, 55(1), 25-35.
>
> [4] Wixted, J. T., & Ebbesen, E. B. (1997). Genuine power curves in forgetting: A quantitative analysis of individual subject forgetting functions. Memory & cognition, 25, 731-739.

---

### Author Response · Authors · 2023-11-22
**Summary of Changes and Significance**

We thank the reviewers for appreciating that our “novel” (*j4qa*) and “comprehensive” (*k3Pn*) approach “advances knowledge tracing” (*rAUs*), has “technically strong” (*rAUs*) and comprehensive design/methodology (*k3pn*/*rAUs*), with impressive/superior results (*j4qa*/*k3pn*, *rAUs*) that “exceed baselines” (*da66*) with extensive validations from multiple angles (*k3Pn*). Additionally, we are glad the reviewers find the manuscript to be clear, intuitive, and well-structured (*rAUs*, *k3Pn*, *j4qa*).

## Summary of changes

Here is a summary of changes we made to address the feedback:
* A new ablation study shows that all components of our model are crucial for its leading performance (summarized in Sec. 4.1 with details in Appendix A.8: Table 16 and Fig. 13).
* We added QIKT and GKT (for a total of now 8 baselines) to strengthen our claims about performance (Figs. 2-3 and Tables 2, 9-12) and interpretability (Fig. 12 and Tables 3, 5, 13, 15)
* We added experiments for the largest cohort size in each dataset (Fig. 2), and find that deep learning baselines require training data from at least 60k learners to reach our predictive performance with just 300 learners (Junyi15 dataset).
* We expanded the interpretability analyses to now include transfer ability and knowledge volatility, successfully relating them to appropriate behavioral measures (Fig. 10 in Appendix A.6.4).
* We clarified the interpretation of cognitive traits (Sec. 3.1), the empirical evaluation of their operational interpretability (Sec. 4.3.1 with improved Fig. 4), the motivation of our choice of datasets (Appendix A.3.2), and the limitations of our datasets and modeling choices (Discussion and Appendix A.3.2).

In summary, these changes now provide even stronger empirical support for our performance and interpretability claims, confirm that all components of our model are needed to achieve superior performance, and improve the clarity of the text reflecting reviewer feedback.

## Significance

In addition to performance, a main contribution of our paper is addressing the interpretability gap, which we agree with reviewer *k3pn* is ”the pain point of the knowledge tracing field”. We believe our work is an important step towards closing this gap, by showing how to build a high-performing model with interpretable representations, and—importantly—providing a comprehensive evaluation framework for interpretability. Our framework covers i) specificity, consistency, and disentanglement of learner representations, ii) graph alignment, and iii) operational interpretability by relating inferred representations to future behavioral outcomes.

This contribution by itself can enrich the knowledge tracing field by advancing the standard for interpretability evaluations. In sum, our work provides an important integration of cognitive science and machine learning (*rAUs*), and we expect our framework to inspire future research and inform practical intelligent tutoring systems (ITS) .

---

### Meta-Review · Area_Chair_A3Bu · 2023-12-08

**Metareview:**

This paper proposes a novel generative modeling approach to the knowledge tracing problem. The combination of a sophisticated model based on Bayesian deep learning with a graphical representation of knowledge relations, which achieves both predictive accuracy and interpretability, is a significant technical contribution.
Several doubts raised by the reviewers, such as lack of explanations and experiments, have been adequately resolved by the author responses.

**Justification For Why Not Higher Score:**

It is a rather complex modeling that delves into the specific nature of the target domain and is somewhat less transferable to other fields. It may not be of much benefit to anyone other than those specifically interested in the target problem.

**Justification For Why Not Lower Score:**

It is technically well-designed and will be welcomed with technical interest not only in the learning analytics field, but also in the ICLR community.

---

### Decision · Program_Chairs · 2024-01-16

Accept (spotlight)